# Assessing Trifecta Achievement after Percutaneous Cryoablation of Small Renal Masses: Results from a Multi-Institutional Collaboration

**DOI:** 10.3390/medicina58081041

**Published:** 2022-08-03

**Authors:** Andrea Piasentin, Francesco Claps, Tommaso Silvestri, Giacomo Rebez, Fabio Traunero, Maria Carmen Mir, Michele Rizzo, Antonio Celia, Calogero Cicero, Martina Urbani, Luca Balestreri, Lisa Pola, Fulvio Laganà, Stefano Cernic, Maria Assunta Cova, Michele Bertolotto, Carlo Trombetta, Giovanni Liguori, Nicola Pavan

**Affiliations:** 1Urology Clinic, Department of Medical, Surgical and Health Sciences, University of Trieste, Cattinara Hospital, Strada di Fiume 447, 34149 Trieste, Italy; andrea.piasentin23@gmail.com (A.P.); giacomorebez@gmail.com (G.R.); fabio.tra92@gmail.com (F.T.); mik.rizzo@gmail.com (M.R.); trombcar@units.it (C.T.); gioliguori33@gmail.com (G.L.); nicpavan@gmail.com (N.P.); 2Department of Urology, San Bassiano Hospital, 36061 Bassano del Grappa, Italy; tommaso.silve@gmail.com (T.S.); antoniocel70@yahoo.it (A.C.); 3Department of Urology, Fundacion Instituto Valenciano Oncologia, FIVO, 46009 Valencia, Spain; mirmare@yahoo.es; 4Radiology Department, San Bassiano Hospital, ULSS 7 Pedemontana, 36061 Bassano del Grappa, Italy; calogero.cicero@aulss7.veneto.it; 5Radiology Department, Centro di Riferimento Oncologico (CRO), 33081 Aviano, Italy; murbani@cro.it (M.U.); lbalestreri@cro.it (L.B.); 6Department of Urology, ULSS 3 Serenissima, 30031 Dolo, Italy; lisa.pola@aulss3.veneto.it (L.P.); fulvio.lagana@aulss3.veneto.it (F.L.); 7Radiology Department, Azienda Universitaria Giuliano Isontina (ASUGI), Università degli Studi di Trieste, 34127 Trieste, Italy; stefanocernic@gmail.com (S.C.); m.cova@fmc.units.it (M.A.C.); bertolot@units.it (M.B.)

**Keywords:** cryoablation, renal cancer, small renal masses

## Abstract

*Background and Objectives*: To assess efficacy and safety of Percutaneous Cryoablation (PCA) of small renal masses (SRMs) using Trifecta outcomes in a large cohort of patients who were not eligible for surgery. *Materials and methods*: All PCAs performed in four different centers between September 2009 and September 2019 were retrospectively evaluated. Patients were divided in two different groups depending on masses dimensional criteria: Group-A: diameter ≤ 25 mm and Group-B: diameter > 25 mm. Complications rates were reported and classified according to the Clavien–Dindo system. The estimate glomerular filtration rate (eGFR) was calculated before PCA and during follow-up schedule. Every patient received a Contrast Enhanced Ultrasound (CEUS) evaluation on the first postoperative day. Radiological follow-up was taken at 3, 6, and 12 months for the first year, then yearly. Radiological recurrence was defined as a contrast enhancement persistence and was reported in the study. Finally, Trifecta outcome, which included complications, RFS, and preservation of eGFR class, was calculated for every procedure at a median follow-up of 32 months. *Results*: The median age of the patients was 74 years. Group-A included 200 procedures while Group-B included 140. Seventy-eight patients were eligible for Trifecta evaluation. Trifecta was achieved in 69.6% of procedures in Group-A, 40.6% in Group-B (*p* = 0.02). We observed an increased rate of complication in Group-B (13.0% vs. 28.6; *p* < 0.001). However, 97.5% were <II Clavien–Dindo grade. No differences were found between the two groups regarding eGFR before and after treatment. Further, 24-months RFS rates were respectively 98.0% for Group-A and 92.1% in Group-B, while at 36 months were respectively 94.5% and 87.5% (*p* = 0.08). *Conclusions*: PCA seems to be a safe and effective treatment for SRM but in the need of more strict dimensional criteria to achieve a higher possible success rate.

## 1. Introduction

The spread of imaging techniques has led to an increased number of renal masses incidentally diagnosed. Renal Cell Carcinoma (RCC) is a very heterogeneous disease comprising both small renal masses (SRMs) incidentally discovered at time of abdominal imaging and aggressive disease with de novo metastatic spread [1,2]. Currently, over 50% of renal tumors are staged as cT1N0M0 at diagnosis. Specifically, most of them are cT1a (less than 4 cm), and are nowadays strictly defined as SRMs [3]. However, a considerable part of these masses are benign tumors and most malignant tumors are early-stage, low-grade with low biological potential for aggressive behavior.

Partial Nephrectomy (PN) is the gold standard treatment for SRMs [4,5,6]. However, the introduction of minimally invasive treatments has led to the rise of other therapeutic strategies. Cryoablation is an accepted feasible alternative, although several retrospective studies and systematic reviews have demonstrated the oncological advantage of PN over ablation therapy.

Percutaneous cryoablations (PCA) are gaining an important role, especially in the management of the elderly and comorbid patients with SRM [7,8]. During the last two decades, several experiences reporting outcomes of PCA confirmed its feasibility, safety, and efficacy as a reliable treatment option in such a setting. There is growing evidence sustaining the oncological safety of PCA, which is expanding the indications to all patients with renal masses of <3 cm in maximum diameter [5,9,10]. The objective of our study was to comprehensively evaluate both functional and oncological outcomes using individual patients’ data (IPD) from a multi-institutional cohort of PCA candidates. We finally assess a tailored Trifecta, which is a composite outcome designed to give to the reader an objective point of view of the success of the treatment at 2 years of follow-up.

## 2. Materials and Methods

Patients: Demographic, clinicopathological, and outcome data were retrospectively collected from medical records of 315 consecutive patients who underwent PCA for de novo cT1N0M0 SRM between September 2009 and September 2019. A total of four tertiary referral centers provided the IPD. The study was designed according to national regulations and the principles of the Declaration of Helsinki in accordance with Good Clinical Practice guidelines. Inclusion criteria for PCA were: (1) any cT1 tumor, irrespective of location and cystic type; (2) maintain a prone position for 60 min; (3) patient is considered at high-risk for surgery whether for high complexity of the renal mass or for comorbidities; (4) patient’s choice of PCA over nephron sparing surgery. Patients were counseled about the risks and benefits of PCA. Indication for PCA was further confirmed at an internal multidisciplinary consensus meeting and all procedures were carried out by a dedicated interventional radiologist at each participating site. All patients underwent contrast-enhanced abdominal/pelvis and thorax computed tomography (CT) scans for RCC diagnosis and clinical staging while magnetic resonance imaging (MRI) and abdominal ultrasonography were provided to further refine the evaluation of the index lesion in selected cases. A renal tumor biopsy was routinely conducted when feasible. Variables collected include age, gender, body mass index (BMI), American Society of Anesthesiologists Classification (ASA) score, and estimated glomerular filtration rate (eGFR) according to Modification of Diet in Renal Disease (MDRD) formula. Anatomical parameters and tumor size were evaluated with the preoperative aspects and dimensions used for an anatomical (PADUA) score [11]. The number of cryoprobes and biopsies’ histological responses were reported as specific procedural parameters. Clavien–Dindo Classification (CD) was used to evaluate the perioperative complication rate [12]. Endpoints: Technical success was achieved when the ice-ball included all renal masses and two frizzing cycles were performed. The median diameter of SRMs included in our series was chosen as an arbitrary cut-off to divide the population into two groups: Group-A included all PCA performed on masses with a diameter ≤25 mm, Group-B included those with masses >25 mm. Three major outcomes were individually evaluated: perioperative complications, eGFR variation, and recurrence-free survival (RFS). The number and type of complications were obtained from procedure reports and medical charts. Any event occurring within the procedure and in-hospital stay was considered. eGFR was calculated before the procedure and at every follow-up visit. eGFR variation was defined as the percentage ratio between postoperative eGFR and preoperative eGFR at 24 months. Further, 10% eGFR variation was considered as clinically significant for the analysis. Due to the lack of consensus on the definition of recurrence after thermal ablation treatment for renal cancer, we considered recurrence the presence of contrast enhancement in the ablated area during follow-up imaging [13]. Data for the oncological outcome were collected from radiological reports of every follow-up. Benign masses and those in which biopsy was not taken were excluded from the analysis. A tailored Trifecta, inspired by the one proposed by Gill et al. [14] represents the endpoint of the current analysis as it is a composite outcome measure focusing on three different topics: recurrence-free survival (RFS) rate, kidney function preservation, and rate of perioperative complications. These included all cases without complications (only Clavien–Dindo >II were considered), no recurrences and no chronic kidney disease (CKD) upstaging at any stage during follow-up time. Only malignant-proven biopsies and non-diagnostic biopsies were considered eligible for Trifecta evaluation.

The procedure: All PCAs were performed with the Visual-ICE^®^ System (Galil Medical Inc., Saint Paul, MN, USA). Different types of CryoNeedle™ (Galil Medical Inc., Saint Paul, MN, USA) were used according to the anatomy and morphology of lesions. The diameter of the probe was 1.5 mm-17G. The most commonly used needles were: IceRod^®^ 1.5 mm plus IceSphere^®^ 1.5 mm. PCA was performed after positioning of the CryoNeedle. A double freeze and thawing cycles were performed. Initial freeze cycles were performed for 10 min, at temperatures below −40 °C, interspersed by at least an 8-min active thaw period. PCA was then repeated for an additional 10 min and the cryoprobes were removed after 3 min of active thaw. The number of cryoprobes was chosen to ensure adequate ice-ball coverage (the target was 5–10 mm over the edge of the tumor). Thermocouples in the probe monitored the ablation site temperature. CT images were repeated during the freeze cycles and at the end of the procedure to confirm the adequate ice-ball creation and absence of complications. To the best of our knowledge, no other series described such confirmative imaging after PCA. Follow-up protocol included a contrast-enhancement ultrasound (CEUS) on the first post-procedure day [15]. Postoperative diagnostic imaging was fundamental to detect both possible early asymptomatic complications and the effectiveness of the treatment. The second follow-up was set at three months from the procedure and then after six and twelve months for the first year, yearly thereafter. Each follow-up included a radiological examination (MRI, CT, or CEUS) and a blood test for creatinine and eGFR evaluation. Whenever disease recurrence was detected, patients were re-evaluated within an internal multidisciplinary consensus meeting, and another treatment was further proposed.

Statistical analysis: Categorical variables were reported as percentages, continuous-coded variables were summarized as median and interquartile range (IQR). Normality of the continuous variables was tested using the Shapiro–Wilk test. Patients were stratified into two groups according to median diameter of lesion (≤25 mm and >25 mm) and variables of interest were compared using the Chi-Squared or Fisher Exact test for categorical parameters, and with the Mann–Whitney test or Kruskal–Wallis test for continuous ones. Local recurrence-free survival (LRFS) was defined as the interval between the date of first PCA and the date of the first local recurrence, with censoring at the last examination date for patients who did not have a local recurrence. The median follow-up was computed for censored patients, excluding patients with the events of interest. LRFS was estimated using the Kaplan–Meier approach. All statistical analyses were performed with R software (version 3.5.0). All tests were two-sided with a level of significance set at *p* < 0.05.

## 3. Results

A total of 340 procedures in 315 patients were performed. Further, 332/340 procedures (97.6%) achieved technical success. Median age at treatment was 74 years (IQR 68–79). Thirty-six (11.4%) patients had a solitary kidney. The median ASA score was 3 (2–3) and BMI was 26.2 (23.9–28.4). The median diameter of the renal masses was 25 mm (range 6–53 mm) with a median PADUA score of 8 (range 6–12). The median age was significantly higher in Group-B (*p* < 0.001) together with the PADUA score (*p* < 0.001), ASA score (*p* = 0.04), and the number of cryoprobes used during the procedures (*p* < 0.001). Preoperative serum creatinine and eGFR were comparable between the two groups. Renal mass biopsy was obtained in 303 (89%) procedures: Group-A 172 (86.0%) vs. Group-B 131 (93.6%, *p* = 0.03). RCC was diagnosed in 66.7% of the cases, 23.1% of the masses were benign, and 10.2% were non-diagnostic. Among those diagnosed with RCC, 143 (70.7%) were clear cell carcinoma, 47 (23.2%) were papillary, and 12 (5.9%) were chromophobe; the benign masses were 51 (72.8%) renal oncocytomas, 2 (2.8%) angiomyolipomas, and 17 (24.2%) papillary adenomas. A detailed description of clinicopathological parameters is reported in Table 1.

The complications rates of the 340 PCAs included in the current analysis and stratified by size are presented in Table 2. The overall perioperative complications rate was 20%. In 66 (19.5%) of cases, complications were reported: (54) 81.8% were CD I, (9) 13.6% were CD II, and (3) 4.6% were CD III, respectively, (two cases required radiological angioembolization and one case required laparoscopic evacuation of renal haematoma). CD III complications occurred in clear cell renal carcinoma histotype. An increased rate of overall complications was observed in Group-B (13.0% vs. 28.6%; *p* < 0.001). Procedures burdened with complications had a similar preoperative PADUA score (8.1 vs. 7.9; *p* = 0.35). None of the PCAs required post-procedure salvage surgery. No patients underwent open or laparoscopic surgery during the postoperative hospital stay.

There were no significant differences between the pretreatment and 24-months post-procedure eGFR (*p* = 0.06). A greater decline was observed in Group-B compared to their counterpart (8% vs. 4%). At 24-months follow-up, serum creatinine values were available for 115 procedures: 72 (62.6%) and resulted in ≤10% of renal function loss. No significant differences between the two groups were found (*p* = 0.56) (see Table 3).

Median follow-up was 27.9 months (range 0.03–110.1 months). A total of 189 procedures were included for RFS evaluation: 19 recurrences (9.9%) were reported. Four of them were at first follow-up and may be considered as persistence: in those patients, a second PCA was performed with technical success and no complications. No biopsy of the previous ablated area was obtained.

The overall RFS was 98.3% at 12 months and 95.4% at 24 months. RFS analysis between the two groups showed a 98.0% RFS in Group-A vs. 92.1% in Group-B at 24 months (*p* = 0.08) (Figure 1). No distant metastasis was detected during follow-up. However, in 2/189 cases (1%), a needle tract seeding was diagnosed.

### Trifecta Achievement

Trifecta achievement among eligible PCAs is presented in Table 4. Only 78 procedures were eligible for Trifecta evaluation at the median follow-up time. The Trifecta accomplishment rate was 69.6% in Group-A vs. 40.6% in Group-B (*p* = 0.02). No CKD upstaging was observed in 54 cases (69.2%). Only one patient was upstaged from grade IV to grade V and only two patients (2.5%) were upstaged from grade III to IV. The major driver of Trifecta rates were complication rates (*p* = 0.05).

## 4. Discussion

In this retrospective, multi-institutional analysis, we evaluated Trifecta outcomes in a cohort of consecutive patients undergoing PCA for non-metastatic SRM. We found that PCA in well selected cases was a safe, feasible, and non-burdening treatment considering perioperative complications’ occurrence, renal function impairment, and the risk of local recurrences [16]. The literature widely discussed the incidence of kidney tumors, mainly because of incidental lesions, and has grown with the increase in treatment rates [17], which is currently focused on nephron sparing strategies. PN rate has increased consistently over the years because of the technological development provided by robot-assisted approaches. The development of minimally invasive percutaneous ablative technologies, such as PCA, has occurred in parallel [18,19,20]. Currently PCA, which does not require general anesthesia, is the least invasive treatment for SRM and Trifecta outcome has been widely used to compare nephron sparing techniques [14,21,22,23]. With such prerogatives, our experience of percutaneous renal tumor cryoablation has yielded a first attempt tumor treatment success of 332/340 (98%), with a complication rate of 13% in Group-A and 28% in Group-B (*p* < 0.001). The majority of these complications are asymptomatic and detected with the help of imaging exams performed the day after the procedure [15]. Major complications have been reported in three cases. The slightly higher complication rate of our series compared to the literature [24,25] may be related to our very strict radiologic follow-up protocol. Perinephric hematomas, for example, were diagnosed on the first postoperative day thanks to the CEUS and probably most of them would not be found without it. The complications rate did not differ between the participating centers. Dimensional criterium appears to be relevant in the outcomes of the PCA, the bigger the lesion the higher is the complication rate, sometimes regardless of the preoperative risks’ algorithms. The use of PCA for large size masses should be weighed carefully [5].

The eGFR percentage variation between the two groups was comparable. Aside from the loss of functional nephrons due to the procedure, the renal function variation is the result of several factors, including age, muscle mass, dietary intake, patient’s comorbidities, and the time of the blood sample. A similar multi-institutional experience [26] quantified the loss of renal function after laparoscopic cryoablation, resulting in a statistically significant difference, however no clinical consequences were observed. Similar to our result, no significant difference between pre-PCA eGFR and the one calculated at every follow-up was observed in a large single center experience by Lim et al. [27]. We finally observed that there is no consensus on how to evaluate the eGFR variation, but in most of the studies on PCA, the loss of renal function did not clinically impact the patient follow-up.

A similar follow-up protocol was described in other experiences [28,29]. Since a correct definition of recurrence has not been provided by the major guidelines, we decided to include all the recurrences in the analysis of oncological outcomes, even early relapsing tumors, which may be defined as persistence of disease. For those particular cases, a gold standard imaging has not been defined so far, and pending further evidence, the decision to perform a second treatment should be decided at referral centers within a multidisciplinary consensus. Additionally, the lack of a biopsy of the ablated area is a limitation of our study, considering that it could have ruled out false positive recurrences. The enhancement persistence in the ablated area can last for over a month after the procedure [30], and the lack of evidence on this topic make the management of these patients challenging.

The overall RFS rate was 91%. At 36 months, Group-A presented 95.7% RFS vs. 84% of Group-B. All the recurrences occurred in the ablated area; no contralateral recurrences were detected. Most of them in our series were found during the first 24 months, confirming usefulness of strict controls in the first two years of follow-up [31]. We also observed two metastases after needle puncture dissemination [32].

The oncologic outcomes of our study are consistent with many studies in the literature on PCA. Kim et al. [33] described their experience both with PCA and laparoscopic CA: at 5-years follow-up, the RFS rate was 86.3% in both groups and also tumor size results to be a predictor of recurrence. On the contrary, the study of Goyal et al. [34] showed a better RFS after PCA at 5-years (95.5%), but persistent tumors (13.7%) were excluded from the analysis and no tumor larger than 4 cm was included. Yanagisawa et al. [35] performed a matched analysis to compare PCA to laparoscopic partial nephrectomy (LPN) for cT1 renal tumor: in the PCA group, similar percentages of RFS to our study were described, but they were inferior to the LPN group.

In regards to oncological outcomes, it must be considered that non-diagnostic biopsies were also considered eligible for Trifecta evaluation. Renal mass biopsy is still underutilized by urologists due to its non-diagnostic rates [36,37]. From a future perspective, confocal microscopy would potentially reduce the amount of non-diagnostic renal mass biopsies, encouraging the urologist to incorporate this tool as part of the decision algorithm. Confocal microscopy allows for rapid imaging of fresh tissue samples and its reliability has been already evaluated among urological malignancies [38,39].

Even if these results are encouraging, nowadays they are still not comparable to the gold standard technique, PN. In a large single center study, Bertolo et al. [40] reviewed their data on PNs performed on 278 patients: they report a cumulative incidence for local recurrence of 3.61% at 5-years follow-up. Moreover, they also observed that bigger masses are more likely to relapse. In another multicenter experience on high-complex renal masses, the higher the PADUA score is, the higher the chance of not achieving optimal surgical outcomes [41]. According to our results and these experiences, the multidisciplinary decision of addressing the patient to PCA must rule out the surgical approach, because of comorbidities and tumor complexity, and considering that bigger masses are more likely to have a local recurrence.

Trifecta outcome was accomplished in 68.0% of Group-A patients compared to 40.6% of Group-B (*p* = 0.02), confirming that the tumor size is very likely to influence the management of the cT1 localized renal tumor undergoing percutaneous treatment. In the original Trifecta study by Gill et al. [14], the authors used this composite outcome to confirm the safety of PN: since this improved over time, a similar percentage of the Trifecta to our study was achieved in his last series of PNs. As far as we know, this is the only study of a large multicenter series that aims to evaluate the Trifecta outcome for PCA. We believe that this composite outcome will give the reader an objective point of view on the safety and reliability of this procedure in such setting.

Our study is not devoid of limitations that must be acknowledged. First, this study was limited by its retrospective nature. Second, the study period included almost ten years, in which different temporal practices and procedure-specific patterns may have existed. Third, we are not able to provide consistent data on hereditary syndromes, which may affect some of our patients with relapsing and multifocal tumors. Renal mass biopsy was not routinely performed. Thus, oncological outcomes might have suffered from such lacking information. Strengths are represented by the multidisciplinary internal consensus meeting for the indications of PCA, a dedicated interventional radiologist performing the procedure at each participating center, and the robust set of IPD extracted from four tertiary referral centers, providing valuable comprehensive evidence for SRMs management.

## 5. Conclusions

In the era of personalized treatment strategies, PCA can achieve a composite outcome “Trifecta” consisting of killing tumoral cells together with preserving renal function and reducing procedural related morbidity. In this multi-institutional series, we found that PCA is more likely to achieve good outcomes measured by Trifecta in masses <2.5 cm than in the bigger one. A major limitation was referred to the high rate of recurrences. In patients not suitable for PN, these results strengthen current indications on PCA in patients with renal masses <3 cm as reported in some recent experiences and could help clinicians during planning and patient counseling.

## Figures and Tables

**Figure 1 medicina-58-01041-f001:**
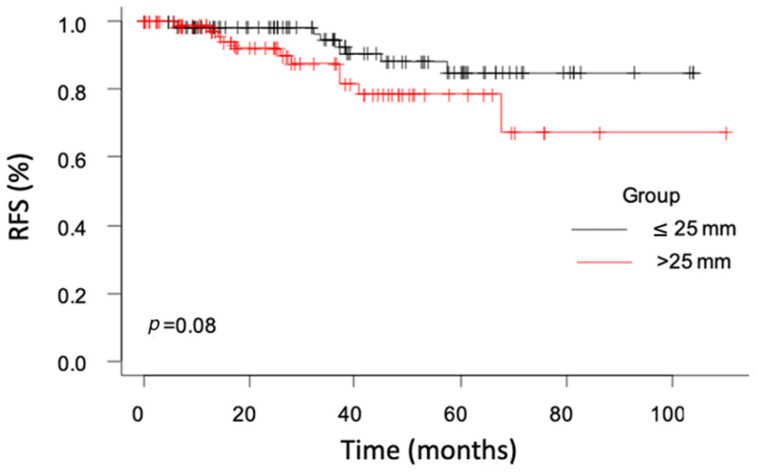
Recurrence-free survival between the two groups.

**Table 1 medicina-58-01041-t001:** Main characteristics of the population, stratified according to the diameter cut-off.

	All Renal Masses (n = 340)	Diameter ≤ 25 mm (n = 200)	Diameter > 25 mm (n = 140)	*p*-Value
Age				
Median (IQR)	74 (67–80)	72 (65.2–78.0)	77 (71–82)	*p* < 0.001
ASA				
Median (IQR)	3 (2–3)	3 (2–3)	3 (2–3)	*p* = 0.04
BMI				
Median (IQR)	26.2 (23.9–28.5)	26.0 (23.7–28.4)	26.4 (24.4–28.7)	*p* = 0.17
Previous RN Number (%)	29 (8.0%)	15 (7.5%)	14 (10%)	*p* = 0.43
Previous PN Number (%)	23 (6.3%)	15 (7.5%)	8 (5.7%)	*p* = 0.66
PADUA SCORE missing 2		Missing 1	Missing 1	
6–7	148 (43.8%)	106 (53.3%)	42 (30.2%)	*p* < 0.001
8–9	141 (41.7%)	77 (38.7%)	64 (46.0%)	
10–11	47 (13.9%)	16 (8.0%)	31 (22.3%)	
≥12	2 (0.6%)	0 (0.0%)	2 (1.4%)	
PADUA SCORE				
Median (IQR)	8 (7–9)	7 (7–8)	8 (7–9)	*p* < 0.001
Creatinine Pre-Treatment				
Mean (SD)	1.2 (0.7)	1.1 (0.6)	1.2 (0.7)	*p* = 0.35
Median (IQR)	1.0 (0.9–1.2)	1.0 (0.9–1.2)	1.0 (0.9–1.3)	
eGFR Pre-Treatment				
Mean (SD)	70.9 (23.9)	72.9 (25.3)	67.8 (21.4)	*p* = 0.15
Median (IQR)	69.9 (56.6–82.6)	73.4 (56.9–84.9)	68.7 (55.5–79.7)	*p* = 0.05
Biopsy				
Yes (%)	303 (89.1)	172 (86.0)	131 (93.6)	*p* = 0.03
No (%)	37 (10.9)	28 (14.0)	9 (6.4)	
Histology				
RCC (%)	202 (66.7)	111 (64.5)	91 (69.5)	*p* = 0.63
Benign (%)	70 (23.1)	43 (25.0)	27 (20.6)	
Non diagnostic (%)	31 (10.2)	18 (10.5)	13 (9.9)	

**Table 2 medicina-58-01041-t002:** Complications occurred in the two groups and the overall population.

Complication	Overall	Diameter ≤ 25 mm	Diameter > 25 mm	*p*-Value
	(n = 340)	(n = 200)	(n= 140)	
Clavien–Dindo				
Clavien I (%)	54 (81.8%)	23 (88.5%)	31 (77.5%)	*p* = 0.12
Clavien II (%)	9 (13.6%)	1 (3.9%)	8 (20.0%)	
Clavien III (%)	3 (4.6%)	2 (7.7%)	1 (2.5%)	
Total				
Number of total complications (%)	66 (19.5%)	26 (13.0%)	40 (28.6%)	*p* < 0.001

**Table 3 medicina-58-01041-t003:** eGFR variation between the two groups.

	Overall Cohort	Diameter ≤ 25 mm	Diameter > 25 mm	*p*-Value
	(n = 115)	(n = 72)	(n = 43)	
Delta EGFR (%)				
Median	6	4	8	*p* = 0.06
(Min–Max)	(−8;6.4)	(−3.2;4.5)	(−8;6.4)	
Delta eGFR (%)				
≤10%	72 (62)	46 (64)	26 (58)	*p* = 0.56
>10%	43 (37)	25 (36)	18 (42)	

**Table 4 medicina-58-01041-t004:** Trifecta outcome between the two groups.

N. Patients	Overall	Diameter ≤ 25 mm	Diameter > 25 mm	*p*-Value
	(n = 78)	(n = 46)	(n = 32)	
Grade eGFR confirmed				
Number (%)	54 (69.2%)	34 (73.9%)	20 (62.5%)	0.28
No Recurrence	71 (91.0%)			
Number (%)		44 (95.7%)	27 (84.4%)	*p* = 0.18
No Complications				
Number (%)	66 (94%)	46 (100%)	28 (87%)	*p* = 0.05
Trifecta				
Number (%)	45 (57.7%)	32 (69.6%)	13 (40.6%)	*p* = 0.02

## Data Availability

The data presented in this study are available on request from the corresponding author. The data are not publicly available because of the monthly enriching of the dataset.

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
