# Peer review of "Assessing Trifecta Achievement after Percutaneous Cryoablation of Small Renal Masses: Results from a Multi-Institutional Collaboration"

_medicina, 2022, doi:10.3390/medicina58081041_

Round 1

Reviewer 1 Report

This manuscript presents very interesting data on percutaneous cryoablation for small renal masses .

I think however that there are a few improvements that should be made before publication.

The authors noted in the manuscript that CKD upstaging occurred in approximately 30%.

What is the percentage of CKD stage 3 or 4 in the study group? Since CKD upstaging is considered to occur in impaired kidney regardless of the procedure, the details should be described.

Although complications were relatively common in this study, only a few cases of G3 complications occurred. What are the details of G3 complications? And did serious complications occur only with benign tumors like AML?

Reviewer 2 Report

The article clarifies the safety and usefulness of cryotherapy, and is clearly and concisely stated and deserves to be published after the modification. The following points should be added in more detail.

1. Cryotherapy is a minimally invasive treatment of choice, which may bias the patient population. For example, patients in poor general condition who cannot undergo general anesthesia or who have undergone repeated surgeries due to VHL genetic abnormalities may be included. It is necessary to clarify whether there is a difference in the background of patients who underwent cryotherapy in the >25mm and <25mm groups.

2. Why are so many benign tumors treated by cryotherapy? Is there a background in which many patients whose malignancy is not actively suspected in the preoperative evaluation are selected for cryotherapy? I am not convinced that there are too many benign tumors. Is there any rationale for an early approach to small tumors that are unlikely to be malignant, even if they are minimally invasive? Can't we just do RAPN when the tumor is a little larger? Cryotherapy has less anti-cancer properties than RAPN, and there is a risk of dissemination, so we should avoid unnecessary use of cryotherapy because it is easy to perform. It is a natural consequence that <2.5mm tumors can achieve trifecta, but I think it is a bit of an overstatement to recommend cryotherapy for <2.5mm. The advantages and disadvantages should be explained to the patient and the patient should be given the choice.

3. Considering that this paper will be read by people other than urologists and radiotherapists, could a little more explanation of trifecta be added?

4. the type of tumor (RCC or not) should also be added to the discussion.

Round 2

Reviewer 2 Report

I believe the manuscript has been sufficiently revised and is worthy of publication.